# A Study of the Effects of the COVID-19 Pandemic on the Experience of Back Pain Reported on Twitter^®^ in the United States: A Natural Language Processing Approach

**DOI:** 10.3390/ijerph18094543

**Published:** 2021-04-25

**Authors:** Krzysztof Fiok, Waldemar Karwowski, Edgar Gutierrez, Maham Saeidi, Awad M. Aljuaid, Mohammad Reza Davahli, Redha Taiar, Tadeusz Marek, Ben D. Sawyer

**Affiliations:** 1Department of Industrial Engineering and Management Systems, University of Central Florida, Orlando, FL 32816, USA; fiok@ucf.edu (K.F.); wkar@ucf.edu (W.K.); edgar.gutierrezfranco@ucf.edu (E.G.); msaeidi@knights.ucf.edu (M.S.); sawyer@ucf.edu (B.D.S.); 2Center for Latin-American Logistics Innovation, LOGyCA, Bogota 110111, Colombia; 3Department of Industrial Engineering, College of Engineering, Taif University, P.O. Box 11099, Taif 21944, Saudi Arabia; amjuaid@tu.edu.sa; 4MATIM, Université de Reims Champagne-Ardenne, 51100 Reims, France; redha.taiar@univ-reims.fr; 5Department of Cognitive Neuroscience and Neuroergonomics, Institute of Applied Psychology, Jagiellonian University, 30-252 Kraków, Poland; marek@uj.edu.pl

**Keywords:** COVID-19 pandemics, back pain reports, Twitter, natural language processing

## Abstract

The COVID-19 pandemic has changed our lifestyles, habits, and daily routine. Some of the impacts of COVID-19 have been widely reported already. However, many effects of the COVID-19 pandemic are still to be discovered. The main objective of this study was to assess the changes in the frequency of reported physical back pain complaints reported during the COVID-19 pandemic. In contrast to other published studies, we target the general population using Twitter as a data source. Specifically, we aim to investigate differences in the number of back pain complaints between the pre-pandemic and during the pandemic. A total of 53,234 and 78,559 tweets were analyzed for November 2019 and November 2020, respectively. Because Twitter users do not always complain explicitly when they tweet about the experience of back pain, we have designed an intelligent filter based on natural language processing (NLP) to automatically classify the examined tweets into the back pain complaining class and other tweets. Analysis of filtered tweets indicated an 84% increase in the back pain complaints reported in November 2020 compared to November 2019. These results might indicate significant changes in lifestyle during the COVID-19 pandemic, including restrictions in daily body movements and reduced exposure to routine physical exercise.

## 1. Introduction

COVID-19 pandemic appeared in early 2020 and quickly spread all over the world [1]. To stop the spread of this very contagious COVID-19 global epidemic, several countries took different actions including social distancing, limiting indoor gatherings, mandating personal protective equipment, and introducing a variety of lockdowns [2,3,4]. As a consequence of the pandemic and related restrictions, the lives of millions of individuals around the world have been changed significantly [5]. Several studies investigated the impacts of the pandemic on the psychological well-being of people at home and at work. For instance, Talevi et al. [6] reviewed fifteen recent studies and observed that many populations suffered from increased mental health problems, with stress and anxiety symptoms reported in the first three weeks of the pandemic. They also showed that some groups—especially females and younger adults—were more vulnerable to such stresses during the pandemic. Mattioli et al. [3] pointed out that quarantine increases anxiety and stress, which can lead to cardiovascular diseases. Furthermore, the close relationship between pandemic and other mental health problems such as self-blame and depression [7], fear [8], and feeling of uncertainty [9] during the COVID-19 outbreak were investigated.

In addition to the aforementioned mental health problems, indoor and outdoor physical activities have also been restricted, which significantly affected people’s everyday lifestyles. Robinson et al. [10] examined the connection between the COVID-19 crisis and obesity through questionnaires with a sample of 2002 participants. The result confirmed that more than 50 percent of participants had no control over their eating habits during the pandemic. Moreover, physical inactivity during quarantine resulted in a variety of musculoskeletal pain, such as back pain, neck pain, and muscle atrophy. Among those, back pain was the most commonly reported type of pain exacerbated by or accompanying the lack of physical activities [11].

As the public discussion on Twitter regarding the COVID-19 outbreak continues, this study aimed to investigate the effect of the pandemic on the frequency of back pain complaints based on publicly available population data contained in Twitter. For this purpose, the following hypotheses were defined as follows:

**Hypothesis** **1** **(H1).***There is a statistically significant difference between the number of complaints regarding back pain during and before the COVID-19 pandemic*.

**Hypothesis** **2** **(H2).***“Back pain” data acquired from those Twitter users who enabled the localization feature in Twitter constitutes a reliable sample of the overall ‘back pain’ tweets*.

In addition to the above, this study investigated changes in the number of actual back pain complaints reported on Twitter over time. For this purpose, the following research steps have been followed: conducting exploratory Twitter data analysis regarding back pain; downloading relevant Twitter data; defining and training an intelligent data filter based on tools from the machine learning (ML), deep learning (DL), and natural language processing (NLP) domain; applying the trained filter to all data instances; creating visualizations of filtered data; testing the research hypotheses.

## 2. Literature Review

### 2.1. Back Pain

Back pain is one of the most common chronic disorders, which not only can reduce work productivity and negatively affect the quality of life [12], but can also increase the economic and societal burden [13]. Furthermore, it is reported that individuals with chronic back pain are worried about losing their jobs and they tend to continue to work without disclosure of pain to their employers [14]. Financial concerns are another reason to take neither sick leave nor therapy facilities [15]. Because of that, in many cases the back pain will not be properly managed. It is reported that for almost 62% of patients with back pain, the pain comes back after one year [13]. The economic burden of back pain is estimated to be between 1% to 2% of the gross national product in Western nations [16]. For example, the economic burden of back pain in the U.S. is calculated at more than 100 billion dollars annually [5].

Due to the lockdown and the necessity to work from home, concerns and complaints of back pain have dramatically emerged. Šagát et al. [5] conducted survey research with 330 participants and found that reports of back pain have increased by more than 11 percent due to the quarantine. Similarly, Pekyavaş and Pekyavas [17] indicated that back pain was the most common pain reported by the participants working at home-offices due to pandemic restrictions. Another research conducted in Turkey during the 3-month pandemic lockdown revealed that the individuals who stayed at home exhibited significantly higher back pain rates than those who continued to work in their offices at their regular workplaces [18]. It should also be noted that physical pain such as headache or lower back pain can be associated with early symptoms of COVID-19 infection [19,20]. Abdullahi et al. [21] conducted a systematic review of 60 articles to assess the evidence on the musculoskeletal and neurological symptoms associated with COVID-19 disease. This review revealed that the COVID-19 patients experienced several musculoskeletal and neurological symptoms, including back pain.

### 2.2. Social Media in Public Health

Surveys and clinical patient data are two major sources of information essential for public health investigations [5,10,17,18]. The advent of social media has introduced a new source of data and has provided the global society a popular communication platform that can be used to examine individuals’ behavior, emotion, and opinions in several domains. The variety and number of studies relying on social media content analysis is growing continuously. For example, Cavazos-Rehg et al. [22] analyzed marijuana-related chats on Twitter to evaluate public attitudes in the context of drug abuse.

In general, Twitter is a valuable asset for enabling the investigation of public health issues during the COVID-19 pandemic. Koh and Liew [23] classified 4492 tweets into three themes based on the level of loneliness, and investigated temporal variations among themes over time. The results showed that tweets do express public sentiments on loneliness during the current pandemic. Sutton et al. [24] analyzed tweets related to lung cancer to better understand cancer communication on Twitter. Lamb et al. [25] reported that the flu infection rate was similar to flu-related to tweets. However, the study also warned that infection tracking could fail if the flu-related tweets do not report an infection. Furthermore, Heaivilin et al. [26] analyzed tweets related to dental pain by using a publicly available dataset. The study used keywords of “dental pain or tooth pain or toothache or tooth ache” and collected a sample of tweets. The results showed that a significant number of tweets related to dental pain are posted every day and they can be easily extracted using simple search terms.

Because of the COVID-19 pandemic and the need for social distancing, the popularity of Twitter has recently increased. Users tend to express their opinions about the pandemic and they like to participate in the public discussion about the Coronavirus. Xue et al. [27] analyzed four million tweets by using ML technique in order to identify popular Twitter discussions and pandemic-related concerns and concluded that COVID-19 related health problems have become one of the common discussions on Twitter since January 2020. Abd-Alrazaq et al. [28] analyzed tweets collected in February and March 2020 and identified several common themes and topics of discussions about COVID-19 pandemic. Mackey et al. [29] examined tweets data using unsupervised machine learning technique to determine COVID-19 related symptoms and individuals’ self-reported experiences related to the infections and their consequences. In addition, Twitter has also been used to better understand public health challenges during the pandemic. For example, Guntuku et al. [30] investigated mental health among Twitter users during COVID-19. After collecting datasets and geolocating all tweets, frequency of single words and phrases were extracted to assess participants’ mental health status. The reported levels of stress, anxiety, and loneliness were higher during the pandemic in comparison with 2019.

## 3. Methodology

This section explains the research methodology, including data, filtering and preprocessing of tweets, and training and testing. The entire process is presented in Figure 1 on a general level and discussed in detail later in this section.

### 3.1. Data Acquisition

The Twitter data acquisition process was carried out in the first days of December 2020. For this study, only English language tweets were considered. We found that obtaining data from Twitter relevant to testing the stated hypotheses can be a challenging task. We have followed Twitter Developer Policy and used official Twitter API to download the necessary data. We have first carried out exploratory analysis to avoid bias in data acquisition process.

#### 3.1.1. Exploratory Twitter Data Analysis

In this study, we analyzed Twitter data for the years 2019 and 2020. The year 2019 served as a baseline before pandemics, and data gathered in 2020 was labeled as COVID-19 related. The initial exploration of Twitter data availability in the selected years was carried out based on the example day of 1 September 2019 and 1 September 2020. From here on, we refer to them as XDay19 and XDay20. The search for term “low back pain” indicated that there were only 134 and 183 tweets in XDay19 and XDay20, respectively. A review of these tweets showed that mostly professionals used the “low back pain” term. Therefore, these tweets were in big portion related to professional discussion, or advertisement, or even counsel, and they did not represent actual back pain complaints. An example of these tweets from Xday19 can be, “*Yoga sessions were related to better back-related function as well as reduced symptoms of chronic low back pain in the biggest U.S. randomized controlled study of yoga so far*”.

We also searched for “lower back pain” which resulted in 164 and 256 for XDay19 and XDay20 respectively. Interestingly, these tweets were mostly user complaints, for instance, “*Today has been rough. Lower back pain, now having bladder spasms. Very uncomfortable. Hope the world can eliminate these shootings. Go to firing range plz...*”.

In the third step, we searched for “back pain” which resulted in a large set of relevant tweets. The number of “back pain” tweets were 3331 and 10,726, correspondingly, for XDay19 and XDay20. Many of these tweets were actual user complaints or related discussion; however, there were also numerous scams, repeated tweets and advertisements. In addition, in some of the selected data, politically related and back-pain-irrelevant tweets were found.

Because of the presence of irrelevant tweets, we tested “back pain” search term with an additional constraint. We searched for two words of the “back” and “pain” and the space between them (“back pain”) in a way that they must appear explicitly together in a tweet, which we refer to as explicit search. This method allowed to obtain 1303 and 5762, correspondingly, for XDay19 and XDay20.

Since we were also interested in testing the hypothesis which required assessing if a similar “back pain” analysis can be constrained to a specific geographical location, we searched for “back pain” tweets with localization set to the USA. By enabling localization, we were able to find 1–2% of all tweets because only this percentage of Twitter users opted-in the localization feature. This search provided 54 and 72 tweets, correspondingly, for XDay19 and XDay20. The summary of our exploratory analysis is presented in Table 1.

#### 3.1.2. Experimental Datasets

Based on the above exploratory analysis, we have decided to conduct two searches: search #1 (S1), that represents “back pain” explicit search with no geographical localization constraint for the months of November 2019 and November 2020, and search #2 (S2) that utilized ‘back pain’ simple search with localization set to the USA over two periods: from 1 March 2019 to 1 December 2019, and from 1 March 2020 to 1 December 2020. The rationale for definition of S1 can be explained as follows. First, choosing a single month instead of the whole year allows to cope with Twitter data download rates and our limited resources. Second, explicit back pain search with no localization allowed to obtain considerable volume of Twitter data with less unrelated data than the non-explicit search. Third, as the lockdown in the USA has begun in March 2020, selecting November 2020 provides us with an opportunity to observe the visible effects of the pandemic on behavioral habits of Twitter users. The rationale behind the S2 is the following. First, selecting the period mentioned above allowed us to test Hypothesis H2 and screen tweets from the whole pandemics period in 2020 and the corresponding period in 2019. Second, as the number of tweets with geographical localization was very low, choosing an explicit ‘back pain’ search would yield an extremely low number of tweets that would not allow carrying out any scientific analysis. A solution of conducting a non-explicit search was introduced to mitigate this issue. Further, to prevent bias caused by tweets that were not relevant to the analysis of back pain in our study, the most irrelevant tweets were filtered according to the methods described in cross-references.

The results of S1 and S2 are as follows. For S1 a total of 53,234 and 78,559 explicit “back pain” tweets were obtained for November 2019 and November 2020, respectively. For S2, a total of 15,663 and 14,634 USA localized tweets were collected for the selected period in years 2019 and 2020, respectively. The screening of downloaded Twitter data indicated that numerous tweets have been repeated and, therefore, contained identical texts. Therefore, we filtered such tweets out by dropping tweets with duplicate texts. The final filtered tweet numbers that were used and analyzed for the purpose of our study are presented in Table 2.

### 3.2. Filtering and Preprocessing of Tweets

Even after removal of repeated tweets, the downloaded Twitter data contained many posts unrelated to back pain or not necessarily expressing complaints regarding the presence of back pain. Therefore, in order to filter out the unwanted tweets and assess the true number of back pain complaints we decided to classify each tweet either as “complaining on back pain” or “other”. Since manual labeling of the total number of 104,274 tweets was not an option due the labor involved, we have developed an automatic filtering method. This method benefits from the recent advantages in the field of ML, DL and NLP and operates on unstructured tweet text only. Based on performance of recent methods in this field [31,32], we decided to train a robustly optimized BERT pretraining approach model (RoBERTa “large” version) [33] and gradient boosting classifier (XGBoost) [34] on a sample of our Twitter data and further automatically infer the classes of all remaining tweets.

The RoBERTa model is a state-of-the-art deep learning transformer model which can be used to output vector representations of text instances. For the best performance of RoBERTa in a given classification task, it is recommended to load a set of parameters pre-trained on a huge textual corpus and fine-tune it in a supervised manner for only a few epochs on the data labeled according to the classification task in question. Afterwards, the high-quality vector representations of the whole text instance can be obtained via the so called CLS (classification) token added to each text instance by the tokenizer used by the RoBERTa model. These vector representations are further used by the XGBoost model, which is a trainable high quality machine learning (ML) classifier capable of outputting a class label for each inputted feature vector.

Since Tweets are written in a specific language, they are sometimes difficult to understand even for humans. Before further analysis and training of the automatic filtering method, tweet texts were preprocessed according to a following procedure:The links to images were replaced with the “_IMAGE” token.Redundant/repeating characters were removed (for example a ten times repeated “a” was converted to “aa”).Textual elements representing retweets were converted to “_RETWEET” token.Other textual elements beginning with “http” or “https” or “bit.ly” or “youtu.be” or “facebook.com” or “instagram.com” were converted to “_URL” tokens.Language of tweets was assessed with use of langdetect module [35] and all non-English tweets were removed.All emoticons were converted to textual representations with use of emoji module [36] (for example “two hearts”).

In order to prepare the data for supervised training of our automatic filtering method we decided to draw a random sample of 5000 tweets from the whole data corpus and labeled all selected tweets manually. Three annotators were asked to separately carry out this task. To assess the extent to which the labels were in agreement we computed the Krippendorff alpha [37] values for the obtained labels. We considered the resulting value of 0.543 to be too low given the binary classification task. Therefore, we repeated the task after an additional clarifying session during which more confusing classification examples have been discussed.

The improved labeling resulted in the higher Krippendorff alpha value of 0.82 which we accepted as sufficient. Because the models training required the obtainment of a single class label from the two possible labels for each data instance, to resolve the remaining disagreements we have assumed that if all three annotators agreed, the chosen label was retained. Furthermore, in the latter case, when two annotators agreed then the label selected by them was also retained. The whole procedure resulted in dividing 5000 tweets into two groups: (1) 2977 tweets with “complaints of back pain”, and (2) 2023 tweets classified as “other”.

### 3.3. Training and Testing

A fivefold cross validation scheme was adopted for training and testing of the developed NLP-based models. Firstly, the feature extraction model RoBERTa was fine-tuned to provide meaningful vector representations (embeddings) of textual data. The model training parameters were adopted from Reference [33] and Reference [32]. Secondly, the XGBoost classifier was provided with the tweet embeddings and trained to carry out the classification. For each cross-validated fold at both training stages, the test sets were retained the same.

To assess the quality of the trained models we have computed the confusion matrix and utilized various classification metrics such as accuracy, balanced accuracy, F1 scores (macro, micro and weighted), Mathews correlation coefficient (MCC), and precision and recall [38,39,40]. However, since the prepared data set was only slightly unbalanced, the differences in obtained values were minor. For the final computation of the quality metrics, we have first concatenated information regarding true values and predictions from five test sets as classified by models trained separately in each fold, and then computed the relevant metrics. After the training and testing procedure was finalized, we have used a single model for inference on the remaining Twitter data. The tweets considered by the model as “other” were dropped before the final analysis of the change of the number of back pain complaining tweets in years 2019 and 2020.

To verify Hypothesis H1, we tested the data from S1 for normality with the use of the Shapiro–Wilk normality test [41] and adopted the significance threshold of 0.05 and later conducted one-way analysis of variance (ANOVA) for tweets with ‘back complaints’ between November 2019 and November 2020. The examined tweets were grouped by days and hours. Similar analysis was carried out for data from S2 with tweets grouped by days. Additionally, in order to prepare for verification of Hypothesis H2, the data originating from S2 was analyzed in a limited time span of November 2019 and November 2020. In the final steps, the data from S1 with November 2019 data and S2 with November 2019 were compared to test the hypothesis H2. This procedure was repeated for year 2020. The level of significance for one-way ANOVA analyses was set to 0.05.

All codes were written in Python version 3.7. The cross-validation procedure and the metrics were implemented with scikit-learn module [42]. RoBERTa model was fine-tuned with use of Flair module [43]. XGBoost classifier was implemented in the XGBoost module [34]. Visualizations were created with matplotlib module [44] after proper data aggregation with use of pandas module [45]. The statistical analysis was carried out with use of scipy package [46]. Twitter data were downloaded with use of searchtweets [47], the official Twitter application for Python, and tweets were parsed with the use of tweet parser module [48]. All experiments were carried out on the same computing machine equipped with a single NVIDIA Titan RTX 24 GB RAM GPU.

## 4. Results

### 4.1. Automatic Filtering Method

The results shown in Table 3 indicate that only 4.56% of all data instances were wrongly classified by the automatic filtering method. Additionally, the results depicted in Table 4 illustrate that the fine-tuned RoBERTa model provided a meaningful vector representations of tweets for the XGBoost classifier.

### 4.2. Hypotheses Testing

After training and testing, we have further used the developed model for the inference of remaining data instances in the downloaded tweet corpus. The results of inferences are displayed in grouped form in Table 5. For S1, it can be seen that the total number of tweets with back pain complaints in November 2020 was much higher and accounted for over 184% of the number of back pain complaining tweets posted in November 2019. A more detailed day-by-day and hour-by-hour analysis is displayed in Figure 2 and Figure 3, which illustrates large differences in daily and hourly number of “back pain” complaining tweets acquired in S1. In both displayed daily and hourly analyses, there is a visible positive trend in November 2020 data, but no such phenomenon is present in November 2019. Additionally, the absolute numbers of complaining tweets are generally higher in November 2020.

Statistical analysis carried out for (1) S1 data grouped by hours was not normally distributed with data from November 2019 characterized by *p* = 2.6858 × 10^−5^ and November 2020 by *p* = 2.8734 × 10^−5^, (2) S1 data grouped by days were normally distributed with data from November 2019 characterized by *p* = 0.406 and November 2020 by *p* = 0.9858, (3) S2 data were normally distributed and the whole analysis period grouped by days in 2019 was characterized by *p* = 0.2016 and in 2020 by *p* = 0.1892, and (4) S2 data, selected Novembers of 2019 and 2020 grouped by days the *p*-value was 0.5311 and 0.6019 correspondingly. For data grouped by days, the differences between number of low back complaints posted in November 2019 and November 2020 were statistically significant (F = 213.4187, *p* < 0.05), confirming Hypothesis H1.

Interestingly, for data originating from S2 other conclusions must be drawn. It appears that in the USA localized data there is no positive trend in both years 2019 and 2020. Additionally, contrary to conclusions from S1, in 2020 the total number of complaining tweets was lower accounting for 96.42% of the tweets posted in year 2019. Statistical analysis of the tweets from S2 grouped by days indicated that the distributions of data were normal. and that the differences in the number of tweets that reported back pain between the periods from 1 March to 1 December for 2019 and the same period for 2020 were statistically significant (F = 5.2608, *p* = 0.022).

Finally, to confirm the suspicion that data and conclusions based on S1 and (2) differ, we approached the H2 hypothesis by comparing data and results for S1 and (2) in comparable time span, i.e., November 2019 and November 2020. Visualization of data extracted according to S2 is visualized in Figure 3. As already mentioned for S1, there were way more back pain complaints in November 2020 than November 2019 andthe differences were statistically significant. However, for S2 limited to the months of November, there were more back pain complaints in November 2019 (1392) than in November 2020 (1275), and no statistically significant difference (F = 2.9181, *p* = 0.093) was observed. Therefore, because analysis of data from S1 and (2) leads to different conclusions, the H2 hypothesis was not confirmed.

## 5. Discussion

As discussed above, this study used two types of Twitter data acquired in S1 “back pain” tweets (explicit search) and in S2 “back pain” tweets with enabled data localization limited to the USA. The unsampled search data provided a full picture of the tweets of interest. We also believe that any analysis of Twitter data limited to tweets with enabled localization should be carried out with caution. This is because in our research we had to reject H2 hypothesis which can be interpreted as a warning, that 1–2% of users who activate the localization feature in Twitter not necessarily constitute a representative group of the whole Twitter community. Therefore, drawing conclusions from data provided only by this group might lead to observations that differ from those obtained based on the analysis of the full data sets. However, we believe that in some cases Twitter data with localization is important source of information since data filtering based on availability of localization allows to identify tweets with very little or no duplication in text content. In our study (see Table 5) less than 1% of downloaded tweets with localization had duplicated texts, suggesting that ‘spam and scam’ accounts do not necessarily turn on the Twitter localization feature.

The results of this study are consistent with previous research related to COVID-19 that analyzed the impact of this pandemic on mental and physical health problems across the world. Furthermore, our findings confirm the general public concerns regarding the adverse health effects of the pandemic, especially those expressed through the social media since early 2020. These include, for example, the published accounts of the COVID-19 epidemic in New York City [49] or lockdown in Turkey [18], as well discussion of the specific mental health problems [6,23]; psychological reactions [7,8,9]; and the effects on obesity and cardiovascular diseases [3,10].

The focus of this study was on assessing the extent of reports of back pain before (2019) and during (2020) the COVID-19 outbreak. Previous studies, such as [5,17,18] reported increase in back pain caused by quarantine and home isolation process. Our findings based on a large sample of Twitter data are in agreement with those studies and suggest a significant (up to 84%) increase in the number of complaints of back pain during the pandemic compared to the immediate pre-pandemic time.

## 6. Conclusions

Obtaining relevant research data from Twitter is not as easy as one might believe. The limit of rates and cost of the data encourages to carrying out exploratory analysis regarding search queries and modifying initial research plans. In this research, to cope with the above constrains, a limited month-to-month analysis was carried out instead of carrying out a full year-to-year comparison. Furthermore, to limit the amount of necessary data, we used filtering to extract only those tweets which have enabled tweets localization. Furthermore, the experiments carried out in S1 of this study were limited to a single month only. It can be expected that the full year-to-year search would provide a more robust picture of the analyzed phenomenon. Additionally, when comparing the number of tweets on a year-by-year basis, one should also keep in mind that there are continuous changes in the number of Twitter users and monthly number of posted tweets. Therefore, to a certain extent, it is possible that the reported differences in the number of back pain complaining tweets could also be attributed to such changes, and not only to the effects of COVID-19 pandemics.

From the computational point of view, the automatic filtering methods based on NLP worked very well, with a total number of wrongly classified tweets lower than 5%. This result was achieved even though the filtering method operated only on unstructured textual data and did not leverage any tweet metadata, which is known to improve prediction performance [50,51]. From this, we can conclude that the previously tested tweet text preprocessing techniques and methods for tweet classification are indeed capable of providing high quality results. While the applied trained automatic filter utilizing DL and ML methods resulted in high predictive performance, some error remains and should be also taken into consideration in the future research.

In this study, Twitter data was collected before (2019) and during the COVID-19 pandemic (2020). By combining the RoBERTa model as a technique for extracting meaningful features from text and gradient boosting process as a machine learning classifier, we effectively classified a sample of tweets posted by individuals who complained of back pain and those who did not. This study indicates that discussions on Twitter can be used to estimate the number of users complaining of back pain in their everyday lives, including during the COVID-19 pandemic. The study results indicate that back pain-related complaints significantly increased during the COVID-19 pandemic (2020) compared to the prior year (2019). Our findings suggest that the COVID-19 outbreak may have had a disproportionately increased prevalence of back pain, creating a need for physical therapy in the general public. This study also provides directions for future research about sentiment analysis on Twitter about ergonomics interventions for home office users to reduce the incidence of back pain.

## Figures and Tables

**Figure 1 ijerph-18-04543-f001:**
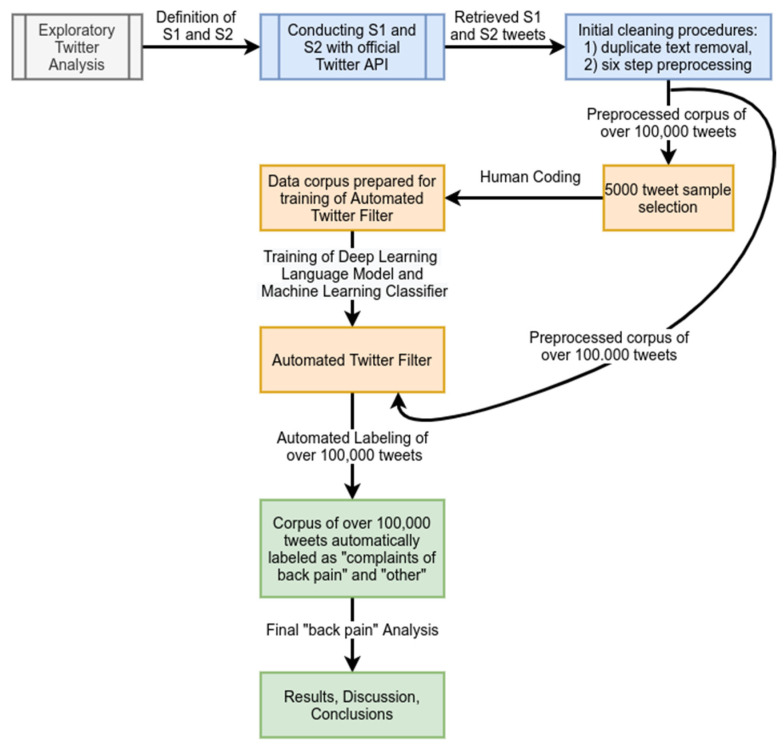
The general-level flowchart of the data analysis process in our study. Detailed descriptions of all visualized steps are available throughout the study. (S1): search #1; (S2) search #2.

**Figure 2 ijerph-18-04543-f002:**
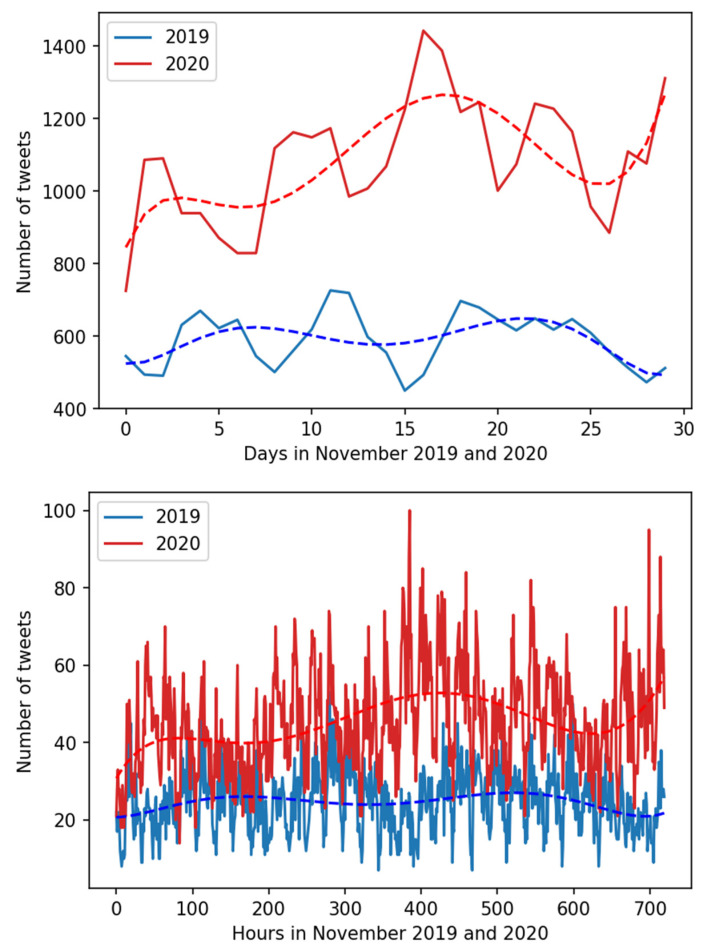
S1 data visualization. Day-by-day and hour-by-hour number of tweets legitimately complaining of back pain. Legend: November 2019 data is in blue, November 2020 data is in red. The dotted lines are trend lines where trend was computed as a 6th order polynomial curve. The colors of the curves match the color of the data plots.

**Figure 3 ijerph-18-04543-f003:**
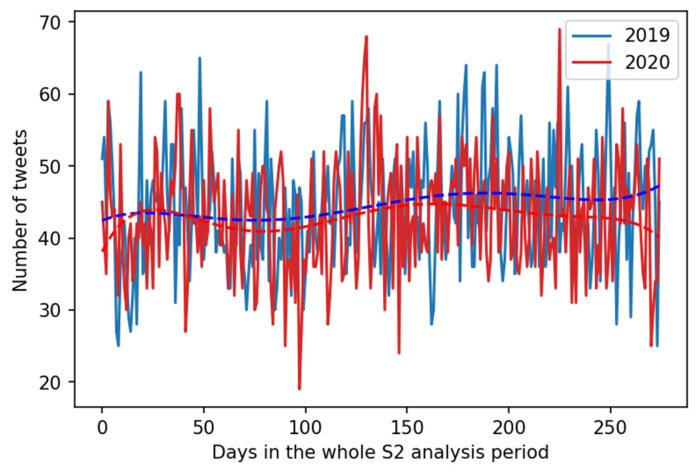
S2 data visualization. Day-by-day number of tweets legitimately complaining about back pain according to Twitter data localized in the USA. Legend: 2019 data is in blue and 2020 data in red. The dotted lines are trend lines where trend was computed as a 6th order polynomial curve. The colors of the curves match the color of the data plots.

**Table 1 ijerph-18-04543-t001:** Summary of the exploratory analysis.

Search Parameters			Number of Tweets		Frequent Tweet Types		
Search phrase	Explicit	Localization	XDay19	XDay20	Complaints	Other (advertisement, professional discussion, not relevant)	Scam
low back pain	No	No	134	183	No	Yes	No
lower back pain	No	No	164	256	Yes	No	No
back pain	No	No	3331	10,726	Yes	Numerous	Yes
back pain	Yes	No	1303	5762	Yes	Yes	Yes
back pain	No	Yes	54	72	Yes	Yes	Yes

**Table 2 ijerph-18-04543-t002:** The number of tweets downloaded in two searches before and after removing tweets with duplicated texts.

Search Number	Time Span	Filtering Duplicated Texts	
		Before	After
1	11 November–1 December 2019	53,234	28,598
1	11 November–1 December 2020	78,559	45,544
2	1 March–1 December 2019	15,663	15,635
2	1 March–1 December 2020	14,634	14,497

**Table 3 ijerph-18-04543-t003:** Confusion matrix of the fine-tuned RoBERTa + XGBoost model.

Group	Other	Complaints of Back Pain
Other	1895	128
Complaints of back pain	100	2877

**Table 4 ijerph-18-04543-t004:** Metrics of the fine-tuned RoBERTa + XGBoost model.

Metric	Value
Accuracy	0.9544
Balanced accuracy	0.9516
F1 macro	0.9526
F1 micro	0.9544
F1 weighted	0.9543
MCC	0.9052
Precision	0.9536
Recall	0.9516

**Table 5 ijerph-18-04543-t005:** Tweets legitimately complaining of back pain. The predictions were carried out by the RoBERTa + XGBoost model described in the methods section.

		Filtering Duplicated Texts			Predicted as Legitimate Complaining Tweets		
Search number	Time span	Before	After	% of original tweet number	After	% of original tweet number	% of filtered tweet number
1	1 November–1 December 2019	53,234	28,598	53.72	17,674	33.2	61.8
1	1 November–1 December 2020	78,559	45,544	57.97	32,530	41.41	71.43
2	1 March–1 December 2019	15,663	15,635	99.82	12,223	78.04	78.18
2	1 March–1 December 2020	14,634	14,497	99.06	11,785	80.53	81.29

## Data Availability

The data presented in this study are available on request from the corresponding author.

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
