# Peer review of "A Study of the Effects of the COVID-19 Pandemic on the Experience of Back Pain Reported on Twitter® in the United States: A Natural Language Processing Approach"

_ijerph, 2021, doi:10.3390/ijerph18094543_

Round 1

Reviewer 1 Report

Dear Editor/Authors,

I read with many difficult the paper entitled "A study of the effects of the COVID-19 pandemic on the experience of back pain reported on TwitterR in the United States: A 3 natural language processing approach". This article demonstrates significant changes in lifestyle during the COVID-19 pandemic related to body movements and time of routine physical exercise.

The work was low interesting, in fact authors compared 53,234 and 78,559 tweets for November 2019 and November 2020, respectively, but it was not clear the method and the results section.

Author Response

Row

Reviewer

Comment

Revision

1

1

I read with many difficult the paper entitled "A study of the effects of the COVID-19 pandemic on the experience of back pain reported on TwitterR in the United States: A 3 natural language processing approach". This article demonstrates significant changes in lifestyle during the COVID-19 pandemic related to body movements and time of routine physical exercise.

The work was low interesting, in fact authors compared 53,234 and 78,559 tweets for November 2019 and November 2020, respectively, but it was not clear the method and the results section.

We appreciate the Reviewer's comments and hope that our research will be positively received by the readers in the context of understanding the effect of COVID_19 on self-reported back pain through Twitter.

We would like to note that Table 2 of our manuscript defines two searches we have conducted, time spans, and numbers of tweets used in our analysis.

Reviewer 2 Report

The manuscript reported change of prevalence of back pain between pandemic and pre-pandemic period, based on machine learning analysis of Twitter data. Authors have done extensive literature review. The results are somewhat expected in that there is significant increase in the number of tweets complaining back pain during pandemic period compared to pre-pandemic period. The topic is interesting and the results may encourage people to change their behaviour by, for example, participating in outdoor activities.

However, I would like the following issues addressed before publication.

1. My understanding is that authors carried out two types of searches, S1 for 'back pain' explicit search, S2 for inexplicit back pain search limited to users enabled localization. And the two searches seemed to yield conflicting results: in S1 there is significant increase in back pain reported during pandemic than pre-pandemic, whereas in S2 this did not seem to be the case.

My concern is that S2 has more than one factors potentially influencing the results. It could be due to geolocation limit or inexplicit search. The latter was not discussed and I wonder if the H2 would hold if authors had used explicit search in S2.

2. Line 66: I don't quite understand why location is needed.  Aren't those tweets whose user locations are disabled unreliable? Many people tend to disable it for privacy concerns, but their tweets are genuine.

3. The authors chose example days of 01-Sep-2019 and 01-Sep-2020, which may not necessarily be representative, for example, if it was a weekend, the number of back pain tweets is presumably less than when it was weekday.

Minor issues

  1. Line 143: Do the authors mean December 2019?
  2. I would simply use 'S1' instead of the verbose 'search #1'
  3. I would suggest add a flowchart representing the steps carried out for this analysis, from preprocessing, cleaning and spliting the data to training machine learning models so that readers can follow more easily and clearly.

  4. Table 3: The group names are better consistent with their previous naming which are "back pain" and "other"
  5.  Line 319: add the test used and p-value
  6. Figures 1 and 2 lack informative description of axes, should be properly labled and stated clearly in the legends

Author Response

Row

Reviewer

Comment

Revision

2

2

The manuscript reported change of prevalence of back pain between pandemic and pre-pandemic period, based on machine learning analysis of Twitter data. Authors have done extensive literature review. The results are somewhat expected in that there is significant increase in the number of tweets complaining back pain during pandemic period compared to pre-pandemic period. The topic is interesting and the results may encourage people to change their behaviour by, for example, participating in outdoor activities.

However, I would like the following issues addressed before publication.

We would like to thank the Reviewer for these kind words.

3

2

1. My understanding is that authors carried out two types of searches, S1 for 'back pain' explicit search, S2 for inexplicit back pain search limited to users enabled localization. And the two searches seemed to yield conflicting results: in S1 there is significant increase in back pain reported during pandemic than pre-pandemic, whereas in S2 this did not seem to be the case.

My concern is that S2 has more than one factors potentially influencing the results. It could be due to geolocation limit or inexplicit search. The latter was not discussed and I wonder if the H2 would hold if authors had used explicit search in S2.

We would like to thank the Reviewer for these kind words.

The Reviewer understands correctly.

The problem with S2 is that if we chose to change only a single factor from S1, i.e., adding geolocation constraint, we would be able to gather an extremely low number of tweets with the explicit search “back pain”. This extremely low number would make any scientific analysis impossible. Therefore, in the struggle to address the problem to any extent, we have admitted the ‘standard search’ meaning that tweets were filtered for words ‘back’ and ‘pain’ but not necessarily standing next to each other. Later, to preserve only tweets meaningful for our analysis, we have used our automated filtering method to exclude unwanted tweets from S2.

The other possibility of simply allowing non-explicit search without geo-localization results in an extremely high number of tweets gathered, which very quickly exceeds the budget of our research in terms of payments required by Twitter to download the required data.

To address this issue in the manuscript, we have modified a fragment of section 3.1.2 (previously lines 196-201), where the rationale for S2 is discussed.

Page 5 line 200-208

4

2

2. Line 66: I don't quite understand why location is needed.  Aren't those tweets whose user locations are disabled unreliable? Many people tend to disable it for privacy concerns, but their tweets are genuine.

Enabling or disabling geolocation wasn't analyzed by us from the point of view of reliability or user credibility. Instead, we were interested if carrying out a similar analysis of back pain yet limited to the USA is feasible, i.e., if there is sufficient data and if the results obtained from such analysis would align with non-geo-restricted analysis. If there was sufficient data and the results would align, it would allow us to carry out a more detailed analysis, maybe even state-by-state. Unfortunately, as demonstrated in our research, this is not the case.

5

2

3. The authors chose example days of 01-Sep-2019 and 01-Sep-2020, which may not necessarily be representative, for example, if it was a weekend, the number of back pain tweets is presumably less than when it was weekday.

We agree with the Reviewer that choosing a single day as a sample has its limitations. Such a solution was selected due to a tradeoff between lust for knowledge and available resources. Also, we hoped that at least the order of magnitude would be represented in a single-day sample. Based on the final numbers of tweets gathered by S1 and S2 and demonstrated in Table 2, we can say that in our case, our probing approach met our requirements and allowed us to obtain a number of tweets that were fitting our resource availability and allowed for proper analysis.

6

2

Minor issues

Line 143: Do the authors mean December 2019?

The manuscript states correctly, all the data was truly acquired in December 2020.

7

2

I would simply use 'S1' instead of the verbose 'search #1'

Thank you for this remark; we have followed your suggestion accordingly.

8

2

I would suggest add a flowchart representing the steps carried out for this analysis, from preprocessing, cleaning and spliting the data to training machine learning models so that readers can follow more easily and clearly.

A flowchart was added according to the Reviewers proposal. It can be found in the manuscript as figure 1 and is presented at the beginning of section 3 along with an introductory sentence.

Page 4 line 143

9

2

Table 3: The group names are better consistent with their previous naming which are "back pain" and "other"

Thank you for this remark, the table was modified as suggested.

Page 8 line 313

10

2

Line 319: add the test used and p-value

The required information is now added to the manuscript. The name of the normality test and adopted significance threshold was added to section 3.3. whereas the obtained p values were added in section 4.2.

Page 7 line 288

Page 8 line 325-331

11

2

Figures 1 and 2 lack informative description of axes, should be properly labled and stated clearly in the legends

We have modified the figures as proposed. Also, the descriptions of figures were enriched with clear S1 and S2 statements regarding the data.

Page 9 lines 337-340 and page 10 lines 358-362

Reviewer 3 Report

Please review the spelling as well as some minor "the"/"a" confusions.

Line

Comment

47

Replace “In addition” with “Furthermore”. (“In addition” is repeated on line 50)

47

Change “disease” with “diseases” (plural).

54

Add “the” after “during”.

55

Replace  “in addition” with “Moreover”; see above.

56

Change “muscular pain” into “muscular pains” (plural).

60

Replace “As a public discussion” with “As the public discussion”.

66

Add “the” before “localization”.

68

Delete “also”. (It is comprised in “in addition” at the beginning of the sentence.)

78

Delete “it” (before “increases”).

82

Add “refer to” before “therapy facilities”.

85

Write “Western” (with capital W) instead of “western”.

86

Replace “estimated to be” with “calculated at”.

88

Add “the” before “necessity”.

93

Add “the” before “3-month”.

95

Replace “that” with “than”.

94

“Individuals who stayed at home” OR “individuals who worked from home”? If the second version, please write in line 95 “those who continued to work in their offices/at their regular workplaces”.

97

Delete “an” in front of “early symptoms”.

109-110

The sentence “Martin et al. [23] 109 analyzed tweets to predict human activity and its location in Spain.” Does not make sense. Something is missing. Please reformulate.

115

Add “to” after “related”.

137

Add a space between “participants’” and “mental”.

141

Replace “and training and testing” with “as well as training and testing”.

145

Replace “test” with “testing”.

147

Add “the” before “necessary”.

150

Replace “two” with “the”.

153

Replace “an example” with “the example”.

153

Add full-stop after “2020”. Continue with Capital F in “from“.

154

Delete the “)” after “XDay20”.

177

Replace “Sicne” with “Since”.

189

Replace TWICE “Decmeber” with “December”.

195

Add “with” after “provide us”.

196

Replace “as follow.” with “threefold:”.

314-315

Replace “illustarets” with “illustrates”.

335

Replace “March 1st” with “March 1st”.

344

Delete one full-stop after “observed”.

373

Change “disease” into “diseases”. (plural) 

399

Delete space after “[50,51]”

435

Add a full-stop after “Lancet”.

Author Response

12

3

Replace “In addition” with “Furthermore”. (“In addition” is repeated on line 50)

Thank you for such a thorough review.

We have modified the indicated error accordingly.

13

3

47

Change “disease” with “diseases” (plural).

54

Add “the” after “during”.

55

Replace  “in addition” with “Moreover”; see above.

56

Change “muscular pain” into “muscular pains” (plural).

60

Replace “As a public discussion” with “As the public discussion”.

66

Add “the” before “localization”.

68

Delete “also”. (It is comprised in “in addition” at the beginning of the sentence.)

78

Delete “it” (before “increases”).

82

Add “refer to” before “therapy facilities”.

85

Write “Western” (with capital W) instead of “western”.

86

Replace “estimated to be” with “calculated at”.

88

Add “the” before “necessity”.

93

Add “the” before “3-month”.

95

Replace “that” with “than”.

94

“Individuals who stayed at home” OR “individuals who worked from home”? If the second version, please write in line 95 “those who continued to work in their offices/at their regular workplaces”.

97

Delete “an” in front of “early symptoms”.

109-110

The sentence “Martin et al. [23] 109 analyzed tweets to predict human activity and its location in Spain.” Does not make sense. Something is missing. Please reformulate.

115

Add “to” after “related”.

137

Add a space between “participants’” and “mental”.

141

Replace “and training and testing” with “as well as training and testing”.

145

Replace “test” with “testing”.

147

Add “the” before “necessary”.

150

Replace “two” with “the”.

153

Replace “an example” with “the example”.

153

Add full-stop after “2020”. Continue with Capital F in “from“.

154

Delete the “)” after “XDay20”.

177

Replace “Sicne” with “Since”.

189

Replace TWICE “Decmeber” with “December”.

195

Add “with” after “provide us”.

196

Replace “as follow.” with “threefold:”.

314-315

Replace “illustarets” with “illustrates”.

335

Replace “March 1st” with “March 1st”.

344

Delete one full-stop after “observed”.

373

Change “disease” into “diseases”. (plural)

399

Delete space after “[50,51]”

435

Add a full-stop after “Lancet”.

Thank you for such a thorough review.

We have modified the numerous errors accordingly.

Regarding Martin et al. [23] (lines 109-110), we removed that reference to avoid misunderstanding.

Reviewer 4 Report

1) Section 2 Objectives is not necessary to be a separate section. please merge section 2 with section 1 introduction.
2) From line 172 to 176, I am confused with the way you test "back pain". If i understand right, there are two ways that you proposed to build the search term: 1) querying two words "back" and "pain" in tweets; 2) query "back pain" in tweets. For the 1st one, could you give several examples? I am not sure if it is appropriate to add the 1st query term. There might be many noise data.

3)In 180, the authors need to describe what are the location features? There are several ways that you can extract location information: 1) location from GPS; 2) user profile; 3) geocoding based on tweet content. In addition, I would like to confirm if you have a real test that 1-2% of all tweets have location features.

4) Could you please explain how do you find Scams, manually or automatically?

5) I suggest authors listing several examples of tweets that are about "not a complaint" and "complaint". How many samples are used for each group? For 0/1 classification, why do you choose XGBoost models instead of logistic regression models or other simple models? I would suggest having a comparative study.  The logistic regression model is easy to test. 

6) In table 4, values should be presented as "0.XXXX"

7) Please have a careful checking on the typos in the manuscript. i.e., line 334, 

8) Please merge section 7 and section 8 into "Conclusion".

Author Response

Row

Reviewer

Comment

Revision

14

4

1) Section 2 Objectives is not necessary to be a separate section. please merge section 2 with section 1 introduction.

Thank you for such a thorough review

We modified text as suggested.

15

4

2) From line 172 to 176, I am confused with the way you test "back pain". If i understand right, there are two ways that you proposed to build the search term: 1) querying two words "back" and "pain" in tweets; 2) query "back pain" in tweets. For the 1st one, could you give several examples? I am not sure if it is appropriate to add the 1st query term. There might be many noise data.

Generally, we ended up with two searches named S1 and S2 with the following characteristics:

S1: "back pain" explicit search with no geographical localization constraint, which means that a tweet was downloaded if it contained two words: 'back' and 'pain' in the given order and separated by a single space. Examples of tweets passing S1: 'Oh i have really bad back pain!!!', 'Back pain? No problem, here is the cure'. Examples of tweets excluded by S1 rules: Uuugh i remember my leg pain back then was really bad!', 'Back !@414m12# pain', 'My back is hurting a lot'.

S2: "back pain" non-explicit search with geographical localization constraint to the USA, which means that a tweet that contains two words: 'back' and 'pain' anywhere in the text and has localization set to the USA. Examples of tweets passing S2: Uuugh i remember my leg pain back then was really bad!', 'Back !@414m12# pain' (both with geolocalization set to the USA), and examples of tweets, not S2: Uuugh i remember my leg pain back then was really bad!', 'Back !@414m12# pain' (without geolocalization), and 'My back is hurting a lot'.

In this manner, in S1 the noisy data was reduced by the fact that the two words have to stand next to each other. In S2, the noisy data was reduced by the requirement of geo-localization set to the USA.

16

4

3)In 180, the authors need to describe what are the location features? There are several ways that you can extract location information: 1) location from GPS; 2) user profile; 3) geocoding based on tweet content. In addition, I would like to confirm if you have a real test that 1-2% of all tweets have location features.

The localization feature was obtained via the official Twitter api. Each Twitter user can declare if she/he agrees to publish his localization while tweeting, so this feature is associated with how the user filled out the user profile. The statistics regarding the 1-2% of localization features availability is taken from the Twitter service documentation

https://developer.twitter.com/en/docs/tutorials/tweet-geo-metadata

accessed 04.06.2021

which states 'The main drawback to relying on Tweet Locations is that only 1-2% of Tweets are geo-tagged.'

17

4

4) Could you please explain how do you find Scams, manually or automatically?

We identified Scams manually during the initial exploratory analysis of downloaded tweets.

18

4

5) I suggest authors listing several examples of tweets that are about "not a complaint" and "complaint". How many samples are used for each group? For 0/1 classification, why do you choose XGBoost models instead of logistic regression models or other simple models? I would suggest having a comparative study.  The logistic regression model is easy to test.

Regarding the machine learning classifier, from our experience, when working with Deep Learning Language Models (DLLMs) and Machine Learning Classifiers, the differences between classifiers were always marginal compared to differences caused by different DLLMs. From our experience, Gradient Boosting classifier performed best most of the time, and that is why we did not consider other machine learning classifiers in this study.

Regarding  the issue of 'how many samples', in lines 261-262 of the manuscript, we report that the number of samples per group was: '1) 2,977 tweets with "complaints of back pain", and 2) 2,023 tweets classified as "other".'

Regarding examples of tweets, please find below the selected examples defining a complaint:

1. 'I have the worst back pain'

2. 'God what is this back pain?'

3. 'Back Pain :weary face:'

4. '@malonezz You should try a inversion table! Relieves so much back pain'

5. 'My back pain is being very stubborn this morning. It’s decided that medication isn’t good enough today.'

, and selected examples defining 'other' tweets:

1. 'Why Suffer with back pain? I often get loads of messages asking what/which exercise can I do to help with my back pain, so I have designed a 4 week Back Care Course to help you. 24 hours before the special offer…  _URL/…'

2. 'Back pain, neck pain treatable with physiotherapy  _URL…'

3. 'Prevent lower back pain by working on this key muscle  _URL…'"

19

4

6) In table 4, values should be presented as "0.XXXX"

We have made the suggested modifications.

20

4

7) Please have a careful checking on the typos in the manuscript. i.e., line 334,

Thank you, we have proofed the manuscript accordingly.

21

4

8) Please merge section 7 and section 8 into "Conclusion".

We have made the suggested modifications.

Round 2

Reviewer 2 Report

Authors have addressed the comments and the manuscript has improved. However, I would suggest some limitations as mentioned in their Response (Row3-5) should be included in the manuscript (as separate Limitations section, or in Discussion)

Line 208: I would suggest cross-reference the section instead of 'below'

Author Response

Thank you for your comment. We replaced "below" with "cross-reference".

Reviewer 4 Report

Thank you so much for your revision. It is well organized and updated in the new manuscript following my suggestions. So, I highly recommend accepting the paper in the current format.

Author Response

Thank you very much.